# Emotional Bias Modification in Youths with Attention Deficit Hyperactivity Disorder (ADHD): New Research Vista

**DOI:** 10.3390/ijerph17114028

**Published:** 2020-06-05

**Authors:** Zhang Melvyn, Aloysius Chow, Ranganath Vallabhajosyula, Daniel SS Fung

**Affiliations:** 1Family Medicine & Primary Care, Lee Kong Chian School of Medicine, Nanyang Technological University, Singapore 308232, Singapore; aloysius.chow@ntu.edu.sg; 2Department of Anatomy, Lee Kong Chian School of Medicine, Nanyang Technological University, Singapore 308232, Singapore; r.vallabhajosyula@ntu.edu.sg; 3Department of Developmental Psychiatry, Institute of Mental Health, Singapore 539747, Singapore; Daniel_fung@imh.com.sg

**Keywords:** emotional bias, cognitive biases, ADHD, child psychiatry

## Abstract

Whilst cognitive bias modification was initially used in the treatment of anxiety disorders, it is also currently being used for the treatment of other psychopathologies. In fact, cognitive bias modification has been especially well-investigated amongst children and adolescents. A recent review suggests some evidence for the modification of interpretative biases amongst children with neurodevelopment disorders. There have since been other studies reporting of the existence of other cognitive biases, such as emotional biases, amongst individuals with attention deficit hyperactivity disorder (ADHD). This perspective article will discuss the epidemiology of ADHD and the nature of emotional biases that are present amongst individuals with ADHD. This perspective article also reviewed some of the studies that have assessed and modified emotional biases in individuals with ADHD. A total of three studies have been identified from the published literature that provide evidence for targeting emotional biases amongst individuals with ADHD. These studies provide us with preliminary evidence for the effectiveness of modifying emotional biases and how it could help in ameliorating symptoms related to emotional dysregulation. There needs to be future research in this area with further evidence supporting the effectiveness of modifying emotional biases. It is also crucial for future research to determine which of these tools is best at detecting such biases, and which of these tools are versatile enough and non-invasive that they could safely be implemented for both research and clinical needs.

## 1. Introduction

Psychological therapies such as cognitive behavioral therapy (CBT) have been the mainstay approach used in the treatment of children and adolescents with psychiatric disorders [1]. To date, CBT is extremely well studied and researched, and it has been included in numerous treatment guidelines globally. In CBT, the therapist typically works with patients to help them to identify and modify cognitions that drive their behavior and the way they feel. Maladaptive cognitions are typically present in individuals with depressive and anxiety-related disorders. The presence of these cognitions results in them favoring stimuli that are disorder-related over neutral stimuli [2]. In the last decade, there have been further advances in experimental psychology thus leading to the discovery of cognitive bias and cognitive bias modification (CBM) [3]. Cognitive bias refers to attentional and approach biases as well as interpretative biases. Attentional bias refers to the preferential tendency of individuals to allocate their attention towards cues, or threatening stimuli, thus ignoring neutral stimuli [4]. Approach bias refers to the automated tendencies of individuals to reach out for stimuli which are of interest [5]. For interpretative biases, they are most common amongst individuals with anxiety disorder and the presence of these biases result in individuals making threatening interpretations of the stimuli presented [6].

Whilst CBM was initially being used in the treatment of anxiety disorders, it is currently also being used for the treatment of other psychopathologies such as depressive disorder [7], substance use disorder [8], obsessive-compulsive disorder [9], and even eating disorders [10]. In fact, cognitive bias modification has been especially well investigated amongst children and adolescents. Krebs et al. [11] examined in their review how modification of negative interpretative biases could negate anxiety symptoms among children and adolescents. They identified a total of 26 studies, with each study including individuals up to the age of 18 years old. They reported that CBM had moderate effects in reducing negative and positive interpretative biases (Hedges’ g was 0.70 and 0.52, respectively) [11]. They also reported that bias modification resulted in a small yet significant reduction of anxiety with an effect size of 0.17 (Hedges’ g) [11]. In another study, Cox et al. [12] explored the utility of CBM in children (38 children) for reducing fears and concerns pertaining to the transition to secondary school education and reported positive effects. It remains clear that cognitive bias modification has clinical utility and is helpful in ameliorating symptoms. It is also evident from these published works that the target for CBM interventions has mainly been interpretative biases. Utilizing a psychological approach makes sense given that pharmacological interventions are typically not considered first for children and adolescents given the side effects of medications.

Most of these prior studies have been conducted amongst typical development individuals with psychopathologies such as anxiety disorder. A recent review by Schmidt et al. [13] sought to determine if cognitive bias modification has been utilized for individuals with neurodevelopmental disorders, such as intellectual impairment, reading and learning difficulties, autism spectrum disorder (ASD) and attention deficit hyperactivity disorder (ADHD). In their review, a total of 29 studies were identified which assessed for cognitive bias modification of interpretative biases. Of these 29 studies, the authors reported that only three studies [14,15,16] included participants with neurodevelopmental disorders and that neurodevelopmental disorders were part of the exclusion criteria for most of the other studies. Of the three studies, one study [14] included 69 participants with mild to borderline intellectual impairments and another study [15] included 75 participants with a mixture of ADHD, ADHD and ASD, and ADHD and oppositional defiant disorder (ODD). In the last study, Sukhodolsky et al. [16] included participants who were aged between 7 to 11 years old and who were referred for anger-related problems. The results from the review suggest that there was some evidence for the modification of interpretative biases amongst individuals with neurodevelopment disorder. Whilst this review is of importance given that it has successfully demonstrated the utility of cognitive bias modification amongst cohorts with neurodevelopmental disorders, it is not without its limitations. Firstly, the studies that have involved participants with neurodevelopment disorders all had participants with multiple comorbidities, hence it is difficult to distinguish the effectiveness of bias modification for a specific neurodevelopmental disorder. Secondly, the review was limited to an examination of the effectiveness of cognitive bias modification for interpretations. There have since been other studies reporting of the existence of other cognitive biases, such as that of emotional biases [17]. For the purposes of this article, we are mainly interested in a specific neurodevelopment disorder, that of ADHD. This perspective article will discuss the epidemiology of ADHD and the nature of emotional biases that are present amongst individuals with ADHD. This perspective article will review some of the studies that have assessed and modified emotional biases amongst individuals with ADHD. It is our intent to perform a review to map out the initial studies that have been done in the field for emotional biases amongst individuals with ADHD. The article will end with a discussion of the future research and clinical implications.

## 2. Epidemiology of ADHD

In this perspective article, the focus is on ADHD given that it is a common neurodevelopmental disorder, with a worldwide prevalence rate of 5% [18]. Individuals afflicted with ADHD typically have a constellation of symptoms, characterized by hyperactivity, impulsivity, and inattention. The presence of these symptoms often leads to significant psychosocial impairments, for example, in academic achievements. Individuals with ADHD often tend to have difficulties with emotional regulation as well [19]. Due to their inherent difficulties with emotional regulation, these individuals tend to be short-tempered and irritable. The advances in experimental psychology has led to the discovery of emotional biases. Targeting emotional biases could potentially help improve the core symptoms of irritability and short-temperedness amongst these individuals. The recent advances in neuroimaging has also revealed that amongst individuals with ADHD, there are reductions in the volume of the bilateral amygdala, nucleus accumbens, and the hippocampus [20]. The presence of these neuroimaging findings might explain the presence of automatic deficits amongst individuals with ADHD [21,22].

## 3. Overview of Emotional Biases

Individuals afflicted with ADHD tend to have difficulties with emotional regulation. Prior research has suggested that emotional biases are present amongst these individuals. Emotional biases refer to the preferential allocation of attention towards emotional stimuli [19]. Emotional biases are found amongst individuals who have the combined subtype of ADHD. Children with ADHD-C typically have difficulties not only in emotional processing, but also in comprehension of others’ emotional state, recognition of facial emotions, matching emotional stories, and orientating towards emotional cues [23]. The theoretical approach suggests these individuals typically have altered top-down processes and top-down processes are typically responsible for executive planning, inhibition, and cognitive control.

## 4. Assessment and Evaluation of Emotional Biases

To identify studies that have been published on emotional biases, articles were identified using a search through the following databases: PubMed, MEDLINE (Ovid), Cochrane Library, PsycINFO and Scopus. A library information specialist from the Lee Kong Chian School of Medicine, Nanyang Technological University Singapore was involved in the refinement of the search strategy. The following search terminologies were used: “cognitive bias modification”, “attention bias modification”, “interpretation bias modification”, “affect processing bias”, “emotional bias” and “ADHD” or “attention deficit disorder”, “attention deficit disorder with hyperactivity”, or “attention deficit hyperactivity disorder”. The search strategy was modified to suit the different databases. The databases were searched from inception through to 18 February 2020. With this, a total of 183 citations were identified, of which 127 were duplicates. In this article, we have decided to review only articles that are in English and if they included (a) children and adolescents with ADHD and (b) mentioned how emotional or affective biases are being assessed or modified. Figure 1 provides an overview of the selection of the studies.

Based on the search strategy, a total of three articles were identified. Pishyareh et al. [23]. undertook the first study examining the visual directions of children with ADHD towards paired emotional scenes. In their research, which involved a total of 30 boys who were of the ages 6 to 11 years old, the visual orientation of these children was tracked using an eye tracking system as they were presented with both emotional and neutral scenes. The authors reported that children with ADHD tend to spend less time on stimuli which were pleasant and tend to attend much more quickly to unpleasant stimuli. Their attentiveness to unpleasant stimuli might in turn result in the symptoms of emotional reactivity. In 2014, Suhulz et al. [24] examined how 14 children with ADHD performed against 14 controls on a face emotion go/no-go task by means of functional magnetic resonance imaging scanning. The functional magnetic resonance imaging demonstrated that children with ADHD tend to have abnormalities in their limbic networks and this could thus explain their difficulties with emotional control. The most recent study was undertaken by Cremone et al. [17]. In their study, 18 children with ADHD were compared with 15 children without ADHD on a dot-probe task and they found positive attention biases among children with ADHD. It is evident from these studies that different paradigms are used in the evaluation of emotional biases. Due to the heterogeneity in the paradigms used, it is difficult for us to synthesize the findings.

## 5. Research and Clinical Implications

These studies provide us with preliminary evidence for the effectiveness of modifying emotional biases and how it could help in ameliorating symptoms related to emotional dysregulation. Emotional biases could therefore be a tangible target in the treatment of children and adolescents with marked emotional dysregulation. A recent meta-analysis by Beheshti, Chavanon, and Christiansen [25] reviewed the evidence of emotional dysregulation in individuals with adult ADHD from 13 studies and reported that individuals with ADHD had significantly higher levels of emotion dysregulation (ED). When looked at more closely, certain aspects of ED (i.e., specifically emotional lability, emotion recognition, and negative emotional responses) were significantly correlated with the severity of ADHD symptoms [25].

Hence, a better understanding of emotional bias modification provides a tool to assess and treat individuals with ADHD to decrease the severity of the ADHD symptoms. However, there needs to be future research in this area, with further evidence supporting the effectiveness of modifying emotional biases. From our knowledge, there has been previous research work targeting other types of cognitive biases amongst individuals with ADHD, primarily that of attentional biases [26,27]. It is believed that individuals with ADHD have underlying impairments in their attentional processes (which is a bottom-up process). It has also been extensively reported that when compared with neurotypical individuals, individuals with ADHD have difficulties with executive functions; one of these functions is broadly defined as inhibition (top-down process), which refers to the ability to inhibit cognitive, motor, verbal, and emotional activities [28]. Due to these underlying impairments, individuals with ADHD have a consequential difficulty with self-control and self-allocation of attention and effort in handling complex cognitive tasks. Previous research has used the visual search paradigm to highlight the deficits that these individuals have. The visual search task typically requires that the individual rapidly scan spatial locations in order to identify a target. Given this, the task helps in the determination of whether individuals could exert control over their attentional processes. It is important to note that whilst targeting attentional processes is of importance in ameliorating difficulties individuals with ADHD have with regards to attention, it does not resolve their difficulties with emotional dysregulation. Thus, targeting emotional biases remains to be of importance and future research should explore this aspect.

From these studies, it is very evident that research into emotional biases is still in its infancy, with only three articles describing how they have investigated for such biases. Nevertheless, these three studies [17,23,24] highlight the fact that children with ADHD tend to have differing responses to emotional stimuli as compared to children without ADHD; and there has been evidence from FMRI studies demonstrating that this might be due to dysfunctions in the limbic circuits. There is thus potential for further research to better characterize these biases and to evaluate if bias modification interventions could help in ameliorating these automatic processes. It is also evident across the three previous studies that diverse tools are being used in the assessment for emotional biases, with tools ranging from eye tracking systems and FMRI to a dot-probe task [17,23,24]. It remains crucial for future research to determine which of these tools is best at detecting such biases and which of these tools are versatile and non-invasive enough to safely be implemented for both research and clinical needs. As aforementioned in the introduction, cognitive bias modification has been investigated previously in adolescents for psychiatric disorders, such as anxiety disorders. It is important for future research to consider the existence of psychiatric comorbidities in individuals with ADHD, as the presence of these comorbidities might influence the effectiveness of emotional bias modification.

## 6. Conclusions

This article has highlighted the potential of assessing and modifying emotional biases, which three prior studies have documented to be present amongst children with ADHD. There is a need for future research in determining the effectiveness of this and in determining the most appropriate tools that could be used for assessment and modification. Targeting emotional biases might help the amelioration of impulsive symptoms in children and children could potentially benefit further if such forms of therapy are combined with conventional psychotherapies.

## Figures and Tables

**Figure 1 ijerph-17-04028-f001:**
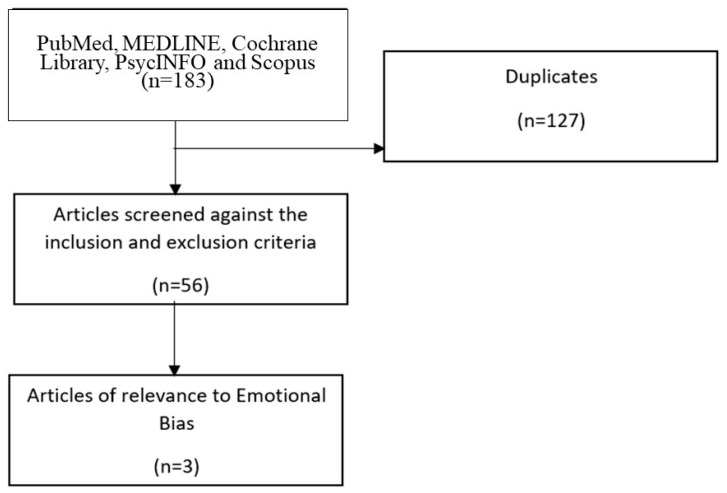
Selection of the studies.

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
