# Peer review of "Emotional Bias Modification in Youths with Attention Deficit Hyperactivity Disorder (ADHD): New Research Vista"

_ijerph, 2020, doi:10.3390/ijerph17114028_

Round 1

Reviewer 1 Report

This commentary made a mini-review on emotional bias modification in youth with ADHD and the directions of further study. It is an important clinical and research issue.

I would like to suggest the authors add some more discussion about the influence of psychiatric comorbidity such as anxiety, depression, oppositional defiant disorder and conduct disorder on emotional bias in ADHD and their influence on intervention.

Author Response

Dear Reviewer,

We thank you for peer reviewing our submitted manuscript. Thank you for recognizing the clinical and the research importance of our work.

We have taken note of your recommendation and have included the following additional statement in the last paragraph of our discussion. The amends are as follows:

“It is important for future research to consider the existence of psychiatric comorbidities in individuals with ADHD, as the presence of these comorbidities might influence the effectiveness of emotional bias modification”

Reviewer 2 Report

Dear Editor,

Thanks for the possibility to review the manuscript titled “

 Emotional Bias Modification in Youths with Attention Deficit Hyperactivity Disorder (ADHD): New Research Vista”. I think that the manuscript addresses an issue that is very important and relevant for the researchers that study the emotions and the neuropsychology of adolescents and adult with ADHD. The objectives are stated clearly and I think that this manuscript can be published with minor changes.

Summary

The authors start introducing literature on cognitive behavioural therapy and then the cognitive bias that refer to both attentional and approach biases, as well as interpretative biases.

Later in the introduction, they show that evidence for the modification of interpretative biases have been applied also to children with neurodevelopment disorder. Later, the authors focus mainly on the nature of emotional biases that are present amongst individuals with ADHD.

They review some of the studies that have assessed for and modified emotional biases among individuals with ADHD and select three studies that provided evidence for targeting emotional biases among individuals with ADHD. The authors underline the preliminary evidence for the effectiveness of modifying emotional biases in ADHD.

The perspective article concludes with needs for further evidence supporting the effectiveness of modifying emotional biases and the necessity to find tools in detecting such biases.

Main impressions

I think that the study addresses an important issue and it is interesting. This perspective of the modification of interpretative biases in ADHD is new.

Observations

The work has a clear logical progression:

  1. 1. Introduction
  2. Epidemiology of ADHD
  3. Overview of Emotional Biases
  4. Assessment & Evaluation of Emotional Biases
  5. Research and Clinical Implications
  6. Conclusions

Although I am not an English native speaker, I think that the language has to be revised.

I think that in the section “Assessment & Evaluation of Emotional Biases” a Prisma diagram of the steps of the papers selected and excluded will help to enrich the paper.

With reference to the sections “Epidemiology of ADHD” and “Research and Clinical Implications” the authors mainly report that the individuals with ADHD have difficulties with executive functions

I suggest to the authors to integrate literature because new neuroimaging studies have shown that not only prefrontal cortex is affected in children with ADHD but also subcortical brain volume. In the same way new neuropsychological studies have shown that not only executive functions are lacking in ADHD, but also automatic bottom-up processes. I refer to the works of Hoogman, Bralten et al (2017)  and Fabio et al. (2015; 2017).

Both references to the subcortical and automatic processes can be useful to explain the factors that influence the performance on cognitive inhibition, attention, arousal, processing speed and Emotional bias

In the conclusion section I think that it is better to describe the results in the light of both top-down and bottom-up theories.

Hoogman, M., Bralten, J., Hibar, D. P., Mennes, M., Zwiers, M. P., Schweren, L. S. J., . . . Franke, B. (2017). Subcortical brain volume differences in participants with attention deficit hyperactivity disorder in children and adults: A cross-sectional mega-analysis. The Lancet Psychiatry, 4(4), 310-319. doi:10.1016/S2215-0366(17)30049-4

Fabio, R.A. (2017). The study of automatic and controlled processes in ADHD: A reread and a new proposal. Mediterranean Journal of Clinical Psychology, 5, 1-8.

Martino, G., Caprì, T., Castriciano, C., & Fabio, R.A. (2017). Automatic Deficits can lead to executive deficits in ADHD. Mediterranean Journal of Clinical Psychology, 5(3), 1-32.

Fabio, R.A., Castriciano, C., & Rondanini, A. (2015).  ADHD: Auditory and Visual Stimuli in Automatic and Controlled Processes. Journal of Attention Disorders, 19(9), 771-778.

Author Response

 We thank you for peer reviewing our submitted manuscript and for your recommendations. We have amended the manuscript and have appended our comments as follows.

Summary

The authors start introducing literature on cognitive behavioural therapy and then the cognitive bias that refer to both attentional and approach biases, as well as interpretative biases.

Later in the introduction, they show that evidence for the modification of interpretative biases have been applied also to children with neurodevelopment disorder. Later, the authors focus mainly on the nature of emotional biases that are present amongst individuals with ADHD.

They review some of the studies that have assessed for and modified emotional biases among individuals with ADHD and select three studies that provided evidence for targeting emotional biases among individuals with ADHD. The authors underline the preliminary evidence for the effectiveness of modifying emotional biases in ADHD.

The perspective article concludes with needs for further evidence supporting the effectiveness of modifying emotional biases and the necessity to find tools in detecting such biases.

Main impressions

I think that the study addresses an important issue and it is interesting. This perspective of the modification of interpretative biases in ADHD is new.

Observations

The work has a clear logical progression:

  1. 1. Introduction
  2. Epidemiology of ADHD
  3. Overview of Emotional Biases
  4. Assessment & Evaluation of Emotional Biases
  5. Research and Clinical Implications
  6. Conclusions

Although I am not an English native speaker, I think that the language has to be revised.

We have attempted to revise the language of the manuscript.

I think that in the section “Assessment & Evaluation of Emotional Biases” a Prisma diagram of the steps of the papers selected and excluded will help to enrich the paper.

We have included a flowchart documenting the selection of the studies. The amends are reflected in the manuscript.

With reference to the sections “Epidemiology of ADHD” and “Research and Clinical Implications” the authors mainly report that the individuals with ADHD have difficulties with executive functions

We thank you for your suggestions. These articles which you have recommended are of relevance. We have included them. The amends are as follows:

“Targeting emotional biases could potentially help improve the core symptoms of irritability and short-temperedness amongst these individuals. The recent advances in neuroimaging has also revealed that amongst individuals with ADHD, there are reduction in the volume of the bilateral amygdala, nucleus accumbens and the hippocampus (Hoogman M et al., 2017) (27).  The presence of these neuroimaging findings might explain the presence of automatic deficits amongst individuals with ADHD (28,29,30).”

I suggest to the authors to integrate literature because new neuroimaging studies have shown that not only prefrontal cortex is affected in children with ADHD but also subcortical brain volume. In the same way new neuropsychological studies have shown that not only executive functions are lacking in ADHD, but also automatic bottom-up processes. I refer to the works of Hoogman, Bralten et al (2017)  and Fabio et al. (2015; 2017).

Both references to the subcortical and automatic processes can be useful to explain the factors that influence the performance on cognitive inhibition, attention, arousal, processing speed and Emotional bias

In the conclusion section I think that it is better to describe the results in the light of both top-down and bottom-up theories.

We have included your suggestions. The amends are as follows:

“It is believed that individuals with ADHD have underlying impairments in their attentional processes (which is a bottom-up process). It has also been extensively reported that when compared with neurotypical individuals, individuals with ADHD have difficulties with executive functions; one of these functions is broadly defined as inhibition (top-down process), which refers to the ability to inhibit cognitive, motor, verbal and emotional activities (Barkley, 2010) (26).”

Reviewer 3 Report

Corrections

  • Please ensure you use the same reference convention consistently throughout the document; there is a mixture of APA and Harvard formats in the text, with the odd inclusion of author’s initials
  • Use of acronyms would make for better reading; e.g., cognitive behavioural therapy (CBT); functional resonance imaging (fMRI)

Highlights

This perspective article reviews a handful of papers related to emotion bias modification in children and adolescents with ADHD.

Whilst the papers intentions are well considered and make for an interesting topic the execution of the work fall short. I will explain in more detail in the following section.

Suggested amendments

The paper states that the aim of the work is to review the literature on CBT for children and adolescents with ADHD, with a focus on interventions that target emotional biases. However, in the introduction, there is an emphasis on interpretative biases, and though this may be related to emotional biases, this relationship is not fully explained or discussed with related literature. I could not tell is you were referring to interpretative biases of emotion or not. This issue could be dealt with by giving a little more detail in the summaries of the primary literature. In the summary of study 12, for example, what form did the CBT take that led to the successful modification of bias? How was bias measured? How old were the sample? The lack of depth in the literature summaries makes for difficult reading, as the topics of AHD, bias, and CBT become abstract, surface levels concepts in the narrative.

Relatedly, there are sections with too few references. Section 3 states that, “Children with ADHD-C typically have difficulties not only in emotional processing, but also comprehension of others’ emotional state, recognition of facial emotions, matching emotional stories, and orientating towards emotional cues.”  Statements like this must be supported by related work, otherwise the reader loses confidence in the authors narrative.

The decision to include papers only published in English is ill-advised. Systematic reviews should always include literature from different language to avoid publication and cultural-artefact biases. Summary statistics of the pooled sample should also be presented with as much demographic information as possible, to index the work relative to other studies (mean age (sd), task, percentage on medication etc). This should be displayed in an APA style summary table.

Furthermore, if the screening process for the review revealed only three articles I would highly recommend modifying your search strategy. A thorough approach was adopted, but the search terms and use of Boolean operators could have significantly reduced the searches coverage of the extant work.

Lastly, summaries of the studies identified are lacking detail, so the author is left with many questions about the effectiveness of the interventions/empirical investigation, and what the key messages are. For instance, is there a particular paradigm that elicits differences in ADHD and TD samples, and what do performance differences tell us about the nature of ADHD?

Summary

This research requires a lot of work to move forward. I hope my suggestions are insightful and lead to a more detailed revision in the near future.

Author Response

We thank you for peer-reviewing our work. We have taken into consideration your recommendations and have made amends to our manuscript. Please find as enclosed our in-line replies.

  • Please ensure you use the same reference convention consistently throughout the document; there is a mixture of APA and Harvard formats in the text, with the odd inclusion of author’s initials

We have amended the referencing to be consistent with the journal’s requirements.

  • Use of acronyms would make for better reading; e.g., cognitive behavioural therapy (CBT); functional resonance imaging (fMRI)

We have amended this.

Highlights

This perspective article reviews a handful of papers related to emotion bias modification in children and adolescents with ADHD.

Whilst the papers intentions are well considered and make for an interesting topic the execution of the work fall short. I will explain in more detail in the following section.

Suggested amendments

The paper states that the aim of the work is to review the literature on CBT for children and adolescents with ADHD, with a focus on interventions that target emotional biases. However, in the introduction, there is an emphasis on interpretative biases, and though this may be related to emotional biases, this relationship is not fully explained or discussed with related literature.

Dear Reviewer, we have restated and clarified the aims of this paper. Our intents for this paper was in rapidly scoping and mapping out the literature for emotional bias amongst individuals with ADHD. We have cited some references in the first two paragraph about interpretation biases, to justify firstly that cognitive bias modification has been considered for youth and adolescents with psychiatric disorder, and secondly, to state that most of the published works are limited to an evaluation of interpretative biases. We have added on this sentence for better clarity:

“It is also evident from these published works that the target for CBM interventions has mainly be for interpretative biases. Utilizing a psychological approach makes sense, given that pharmacological interventions are typically not considered as first line amongst children and adolescents, given the side effects of medications.”

I could not tell is you were referring to interpretative biases of emotion or not. This issue could be dealt with by giving a little more detail in the summaries of the primary literature. In the summary of study 12, for example, what form did the CBT take that led to the successful modification of bias? How was bias measured? How old were the sample? The lack of depth in the literature summaries makes for difficult reading, as the topics of AHD, bias, and CBT become abstract, surface levels concepts in the narrative.

We have decided to remove Study 12, as we realized that the sampled population were between the ages of 18-30, and not youths and adolescents. We have attempted to provide more information for the primary studies that we have included in the discussion.

Relatedly, there are sections with too few references. Section 3 states that, “Children with ADHD-C typically have difficulties not only in emotional processing, but also comprehension of others’ emotional state, recognition of facial emotions, matching emotional stories, and orientating towards emotional cues.”  Statements like this must be supported by related work, otherwise the reader loses confidence in the authors narrative.

We have appended appropriate references.

The decision to include papers only published in English is ill-advised. Systematic reviews should always include literature from different language to avoid publication and cultural-artefact biases. Summary statistics of the pooled sample should also be presented with as much demographic information as possible, to index the work relative to other studies (mean age (sd), task, percentage on medication etc). This should be displayed in an APA style summary table.

We seek to clarify that the intention of this paper is that of a scoping literature review and not that of a full systematic review. A scoping review makes sense, given that this area which we are exploring (emotional biases amongst individuals with ADHD) is a novel area, and there is to be expected only several publications. A scoping review will help us map out the initial work that has been done. We have summarized the results descriptively. We have clarified this by modifying the aims which we have stated.

The amends are as follows:

“This perspective article will also scope and review some of the studies that have assessed for and modified emotional biases amongst individuals with ADHD. It is our intent to perform a scoping review to map out the initial studies that have been done in the field for emotional biases amongst individuals with ADHD. The article will end with a discussion of the future research and clinical implications.”

Furthermore, if the screening process for the review revealed only three articles I would highly recommend modifying your search strategy. A thorough approach was adopted, but the search terms and use of Boolean operators could have significantly reduced the searches coverage of the extant work.

We have worked with a library information specialist from the Lee Kong Chian School of Medicine, to refine the terminologies used in the search strategy. We seek to clarify that we have had to use the Boolean operators to refine the search strategy, otherwise, there were too many irrelevant articles.

Lastly, summaries of the studies identified are lacking detail, so the author is left with many questions about the effectiveness of the interventions/empirical investigation, and what the key messages are. For instance, is there a particular paradigm that elicits differences in ADHD and TD samples, and what do performance differences tell us about the nature of ADHD?

We seek to clarify that we have included relevant details for each of the studies, such as the nature of the paradigm used for the evaluation of the biases, and the sample involved and the outcome. However, due to the heterogeneity in the paradigms used, it remains difficult for us in establishing any definitive results.

Summary

This research requires a lot of work to move forward. I hope my suggestions are insightful and lead to a more detailed revision in the near future.

We have made some necessary revisions to better clarify our intended message.

Round 2

Reviewer 3 Report

I would like to thank the authors for their amendments.

The purpose of the paper is now muh clearer. Mission statements about the intention of the scoping review, details of the information specialist, and Figure 1 help to achieve this. Figure 1 in particular emphasises the effort taken to obtain the relevant literature. 

The use of acronyms also makes for better reading.